# Increasing suppression of saccade-related transients along the human visual hierarchy

Tal Golan[1], Ido Davidesco[2], Meir Meshulam[3], David M Groppe[4,5,6], Pierre Mégevand[4,5], Erin M Yeagle[4,5], Matthew S Goldfinger[4,5], Michal Harel[3], Lucia Melloni[7,8], Charles E Schroeder[9,10], Leon Y Deouell[1,11], Ashesh D Mehta[4,5], Rafael Malach[3]*

[1]Edmond and Lily Safra Center for Brain Sciences, The Hebrew University of Jerusalem, Jerusalem, Israel; [2]Department of Psychology, New York University, New York, United States; [3]Department of Neurobiology, Weizmann Institute of Science, Rehovot, Israel; [4]Department of Neurosurgery, Hofstra Northwell School of Medicine, Manhasset, United States; [5]The Feinstein Institute for Medical Research, Manhasset, United States; [6]The Krembil Neuroscience Centre, Toronto, Canada; [7]Department of Neurophysiology, Max Planck Institute for Brain Research, Frankfurt am Main, Germany; [8]NYU Comprehensive Epilepsy Center, Department of Neurology, School of Medicine, New York University, New York, United States; [9]Department of Neurosurgery, Columbia University College of Physicians and Surgeons, New York, United States; [10]Cognitive Neuroscience and Schizophrenia Program, Nathan Kline Institute, Orangeburg, United States; [11]Department of Psychology, The Hebrew University of Jerusalem, Jerusalem, Israel

*For correspondence: rafi. malach@gmail.com

**Abstract** A key hallmark of visual perceptual awareness is robustness to instabilities arising from unnoticeable eye and eyelid movements. In previous human intracranial (iEEG) work (Golan et al., 2016) we found that excitatory broadband high-frequency activity transients, driven by eye blinks, are suppressed in higher-level but not early visual cortex. Here, we utilized the broad anatomical coverage of iEEG recordings in 12 eye-tracked neurosurgical patients to test whether a similar stabilizing mechanism operates following small saccades. We compared saccades ($1.3°-3.7°$) initiated during inspection of large individual visual objects with similarly-sized external stimulus displacements. Early visual cortex sites responded with positive transients to both conditions. In contrast, in both dorsal and ventral higher-level sites the response to saccades (but not to external displacements) was suppressed. These findings indicate that early visual cortex is highly unstable compared to higher-level visual regions which apparently constitute the main target of stabilizing extra-retinal oculomotor influences.
DOI: https://doi.org/10.7554/eLife.27819.001

## Introduction

The disparity between the apparent stability of the perceived image and the rapidly shifting retinal image during saccadic eye movements has long captivated scientific interest (for a review, see *Wurtz, 2008*). A key facet of this perceived visual stability is the marked difference in the subjective perception of an external stimulus displacement compared to a similar retinal shift caused by a saccade, which likely reflects an active, extra-retinal compensation for the saccade (*Thiele et al., 2002*). Apart from its relevance to the understanding of visuo-oculomotor function, this striking discrepancy

between optics and perception offers a unique means of probing whether and how a given neural representation is related to the contents of subjective visual awareness under highly ubiquitous and ecological conditions.

The most investigated candidates for the neural underpinning of the perceptual distinction between external stimulus displacements and saccades are dorsal higher-level visual regions, and in particular the middle temporal (MT/V5) and the middle superior temporal (MST) areas, whose neurons were shown to strongly differentiate between external and saccadic movements (*Thiele et al., 2002*). This response pattern is compatible with the association of MT/MST activity with perceived (versus retinal) motion (*Tootell et al., 1995*; *Zeki et al., 1993*) and was recently linked with input originating from the superior colliculus and arriving in MT through the pulvinar (*Ridder and Tomlinson, 1997*; *Uematsu et al., 2013*). Other players related to this process are the lateral and ventral intraparietal area where remapping effects (*Duhamel et al., 1992*) and head-centered receptive fields (*Duhamel et al., 1997*) were documented.

Early retinotopic visual cortex (V1/V2/V3), in contrast, is usually believed to reflect retinal displacements regardless of whether they were produced by self- or external- motion (e.g., for a recent demonstration see *Meirovithz et al., 2012*). However, this view is not unanimous, and multiple conflicting findings have been reported. Starting with the pioneering work of *Wurtz (1969)*, several groups have argued for similar responses of V1 to external and saccadic stimulus displacements, implying retinal-like coding of motion in V1 (e.g. *Ilg and Thier, 1996*; *Fischer et al., 1981*). Others have reported responses that were mostly compatible with at least some level of extra-retinal modulation (e.g., *Gawne and Martin, 2002*; *Kagan et al., 2008*) and some have argued strongly for differential responses, including very recent work readdressing this old but yet unsettled question (e.g., *Troncoso et al., 2015*; *McFarland et al., 2015*). This discrepancy across studies may be related to differences in the experimental paradigms and data analysis methods employed (*Troncoso et al., 2015*).

Another less discussed potential contributor to perceptual stability is ventral high-level visual cortex (or Inferior Temporal, in non-human primates). This area, characterized by category-selective responses, has traditionally been viewed as having extremely large receptive fields (see review in *Sayres et al., 2010*), and therefore insensitive to both small saccades and small external displacements as long as they are not large enough to replace the objects benefiting from foveal acuity (*Levy et al., 2001*). However, findings indicating a release from adaptation following small external stimulus displacements (*Grill-Spector et al., 1999*) as well as reports of position-dependent responses (*Sayres et al., 2010*), show that much is still unknown regarding whether and how ventral high-level visual cortex handles the rapid stimulus displacements caused by saccades. Recent findings of saccade-related suppression in V4 (*Zanos et al., 2016*) may suggest that higher-level ventral regions might be affected by saccadic efferent copy as well.

Where in the visual cortex does the distinction between external- and saccade-related motion arise? Does it start at early retinotopic processing or does it appear exclusively downstream? While recordings in non-human primates have provided the bulk of existing knowledge on eye-position related visual processing, such studies are typically very focused in their anatomical coverage, targeting one or two areas. This narrow anatomical coverage might veil the greater functional-anatomical context by which saccades are neurally distinguished from external displacements. In particular, studies focused on MT without sampling earlier cortical regions cannot preclude the possibility that the observed effects in MT are inherited from the early retinotopic visual areas. Conversely, observing significant differences between saccades and external displacements in studies focused exclusively on V1 does not rule out the possibility that higher-level regions may display a far greater level of differential processing of saccades compared to external displacements. However, to the best of our knowledge, such a direct electrophysiological comparison of saccades and external displacements across the visual hierarchy has not been made yet.

Human intracranial EEG recordings (ECoG/SEEG), conducted for clinical diagnostic purposes, often sample the cortex with a substantial anatomical coverage, while still accessing population activity at the millimeter/millisecond spatiotemporal scale. We have recently used this technique to address a similar perceived stability issue arising during eye blinks. In that study, we found that in

high-level visual cortex transient responses to visual interruptions were suppressed when such interruptions were produced by blinks, but not when they were produced by external stimulus blanking (*Golan et al., 2016*). In contrast, both kinds of interruptions elicited similar responses in early visual cortex. Here, we tested whether these findings can be generalized to include the distinction between external stimulus displacements and saccades, by contrasting the cortical responses induced by these two events in the experimental sessions previously reported in *Golan et al. (2016)*.

## Results

Twelve subjects undergoing invasive neurophysiological monitoring for clinical indications viewed still images (15.8° wide) of faces, as well as non-face control stimuli (1 s of continuous display per image). Blocks of ten trials (ten consecutive images) were separated by an intermittent gray screen lasting three seconds, serving as a baseline. In some of the blocks (12 out of 44), a red dot was presented at the center of the stimulus, and the subjects were pre-instructed to fixate on it. In the other 32 blocks, no fixation cross was presented, and the subjects viewed the stimuli freely. During the fixation blocks, excluded from analysis in *Golan et al. (2016)*, horizontal image displacements of either 1.3° or 3.7° visual degrees were introduced at latencies of 300, 500 or 700 ms from trial onset. Eye position was monitored by a video eye tracker (EyeLink 1000, SR Research, Ontario, Canada) synchronized with the stimulus-presenting laptop and the iEEG recording station. Offline, saccades verified to be unrelated with eye blinks and within the magnitude range of 1.3° to 3.7° were marked for comparison with the external stimulus displacements. Note that both the external displacements and the selected saccades were considerably smaller than the extent of the stimuli (15.8°). Subjects were instructed to click a mouse button when they saw an animal image, and these trials were excluded from further analysis.

High-frequency broadband activity (HFB, 70–150 Hz) was estimated from the intracranial recordings as a proxy of population mean firing rate (for further details see *Golan et al., 2016*). 115 contacts showed significant and considerable visual response to the images (inclusion criteria: $p < 0.05$ Bonferroni corrected and effect size $\geq 2$ baseline standard deviations). A General Linear Model (GLM) with a Finite Impulse Response (FIR) basis set was used as a means of deconvolving the overlapping neural responses to the appearance of new stimuli, external stimulus displacements, and saccades, estimating the unique contribution of each experimental condition to the observed time-course (see Material and methods). Blinks and gaps were accounted for as well in the FIR GLM model, but their estimates are not reported here (see *Golan et al., 2016*). Saccades smaller than 1.3° or larger than 3.7° were also accounted for using separate predictor sets but are not further analyzed as there were no matching external displacements that would enable a controlled comparison.

### Grand averages of the contribution of external displacements and saccades to HFB

We averaged the deconvolved traces related with selected saccades or external displacements across electrodes and subjects, using seven visual regions of interest (ROIs): V1, V2, V3, V4, VO, face-selective electrodes and high-level non-face selective electrodes (*Figure 1*). As can be readily observed, in early visual areas, both external displacements and saccades produced a subsequent burst of HFB activity, yielding an overall similar activity profile for these two conditions. However, in higher order areas along the visual hierarchy- areas V4, VO, face-selective and non face-selective high-level visual regions, saccades produced almost no HFB response modulation whereas external displacements of the stimulus were still effective in triggering an HFB activity increase.

### Within-electrode testing of external displacements against saccades

To statistically test this apparent divergence of saccade-related and external displacement-related responses, we conducted a random permutation test within each electrode by shuffling saccade and external displacement event-labels 10,000 times, re-estimating their GLM FIR models in each simulation. For each condition (saccades and external displacements), we detected the largest cluster of continuously above zero HFB-response.

The test statistic was defined as the area under the displacements-related cluster minus the area under the saccades-related cluster. This approach does not assume exact temporal registration of saccade and displacement traces, and it minimizes any potential offsetting of positive HFB responses

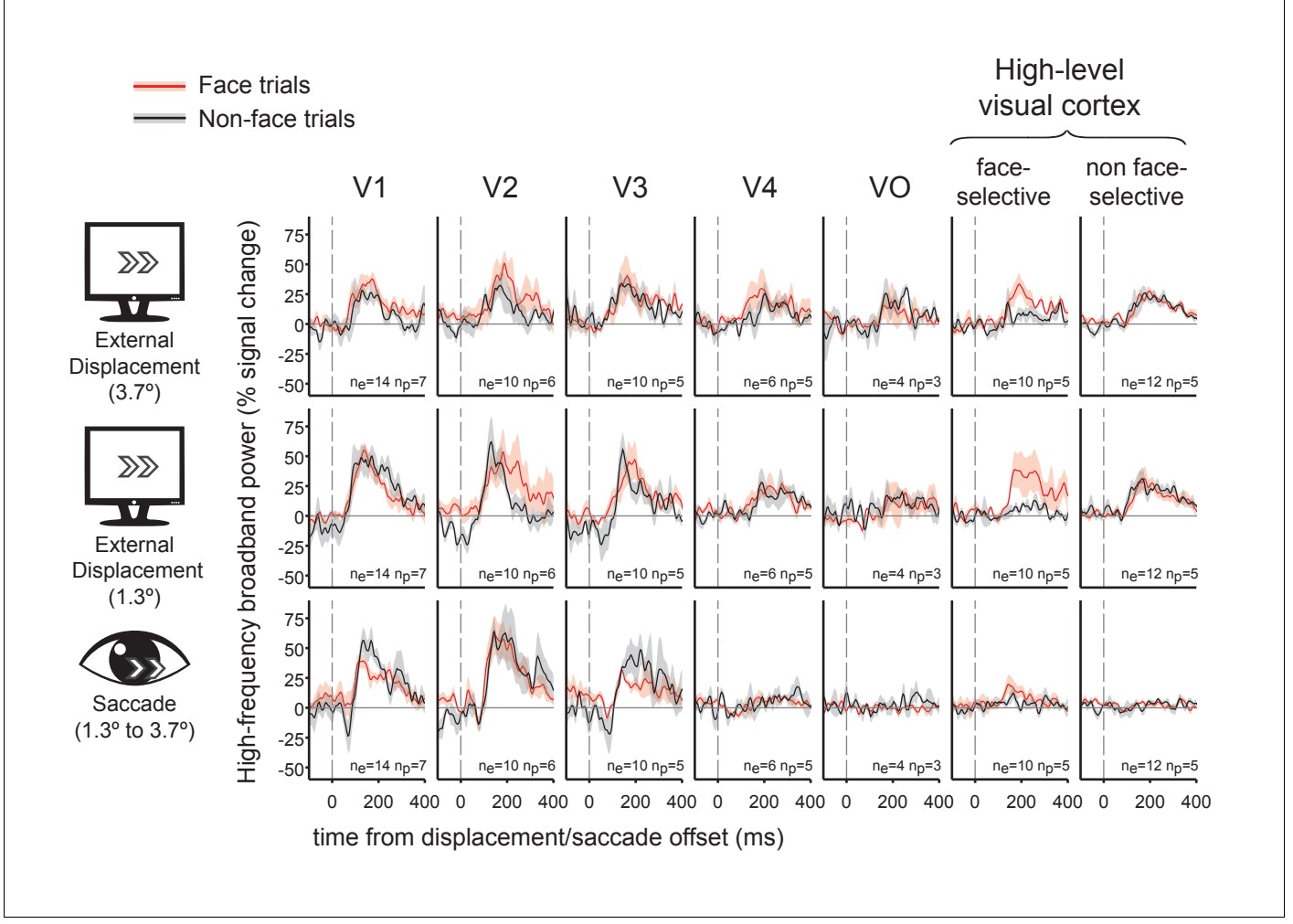

**Figure 1.** Grand-average traces of the contribution of external stimulus displacements (larger and smaller) and saccades to high-frequency broadband activity in seven regions of interest. Shading around the grand average waveforms denotes ±1 standard error (between-subjects error variability). $n_e$ denotes the number of electrodes per trace and $n_p$ denotes the number of subjects per trace. Note the similar HFB activity increase following both saccades and external displacements in V1 to V3. In contrast, V4, VO, face-selective electrodes and non-face selective high-level electrodes all showed highly divergent responses to displacements and saccades. See *Figure 1—figure supplement 1* for latencies and magnitudes of the underlying displacement and saccade events.

DOI: https://doi.org/10.7554/eLife.27819.002

The following figure supplement is available for figure 1:

**Figure supplement 1.** Latencies and magnitudes of external stimulus displacements and saccades.

DOI: https://doi.org/10.7554/eLife.27819.003

by negative ones (in contrast to plain averaging of response traces over time). *Figure 2* presents the results of this test for all electrodes, plotted on a common cortical map, comparing both small and large external displacements to saccades. Multiple sites (36/115) showed significantly greater HFB contribution of external displacements compared to saccades ($p_{FDR} < 0.05$), with greater occurrence of significant difference in visual regions anterior to V3: V1,(1/15), V2 (0/11), V3 (3/12), V4 (4/6), VO (2/6), face-selective electrode (6/15) and high-level non face-selective electrodes (10/18), the latter ROI including both ventral and dorsal high-level sites. The differences showed a significant discrepancy across ROIs (randomization test of independence, $\chi^2(6)=12.64$, p=0.004). No site showed a significantly greater HFB increase for saccades compared with external displacements. Repeating this analysis while using only saccades whose onsets were in the range of 300–700 ms post trial-onset resulted in a highly comparable statistical map (*Figure 2—figure supplement 1*), with 39/115

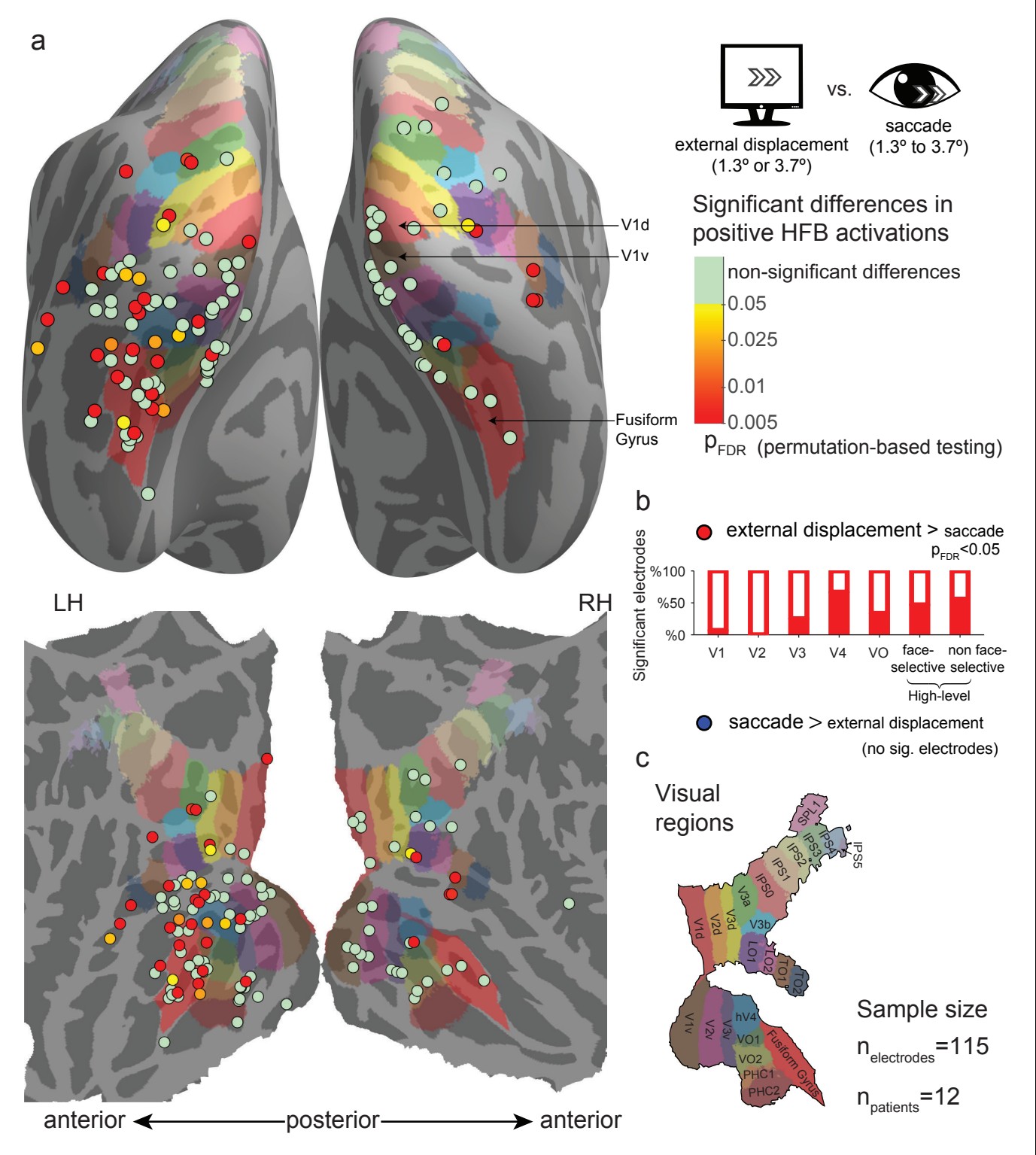

**Figure 2.** Significant differences between positive high-frequency broadband (HFB) activity increase following external stimulus displacements (either 1.3° or 3.7°) and positive HFB activity increase following saccades (1.3° to 3.7°). See *Figure 2—figure supplement 1* for the same analysis using only saccades whose onsets were in the range of 300–700 ms post trial-onset. See *Figure 2—figure supplements 2* and *3* for a comparison of the saccades with only smaller (1.3°) or larger (3.7°) external displacements. (a) Electrode-level statistical map across subjects. Each circle marks the location of one electrode on a common cortical template, either inflated (top) or flatten (bottom). The color code represents FDR-corrected *p* values for electrodes showing significantly greater positive responses to external displacements compared with saccades. None of the visually responsive sites showed

*Figure 2 continued on next page*

*Figure 2 continued*

significantly greater HFB response increases to saccades compared with external displacements. (**b**) The partially filled bars present the percentage of electrodes showing a significantly greater activity increase following external displacements compared with saccades within each region of interest. (**c**) The colored areas on the cortical surface were derived from a surface-based atlas of retinotopic areas (*Wang et al., 2015*) and Destrieux Atlas (*Destrieux et al., 2010*) as implemented in FreeSurfer 5.3 (Fusiform Gyrus, in red).

DOI: https://doi.org/10.7554/eLife.27819.004

The following source data and figure supplements are available for figure 2:

**Source data 1.** Individual electrode data for *Figure 2*.

DOI: https://doi.org/10.7554/eLife.27819.008

**Figure supplement 1.** Significant differences between positive high-frequency broadband (HFB) activity increase following external stimulus displacements (either 1.3° or 3.7°) and positive HFB activity increase following saccades (1.3° to 3.7°), using only saccades whose onsets were in the range of 300–700 ms post trial-onset.

DOI: https://doi.org/10.7554/eLife.27819.005

**Figure supplement 2.** Significant differences between positive high-frequency broadband (HFB) activity increase following small external stimulus displacements (1.3°) and positive HFB activity increase following saccades (1.3° to 3.7°).

DOI: https://doi.org/10.7554/eLife.27819.006

**Figure supplement 3.** Significant differences between positive high-frequency broadband (HFB) activity increase following large external stimulus displacements (3.7°) and positive HFB activity increase following saccades (1.3° to 3.7°).

DOI: https://doi.org/10.7554/eLife.27819.007

---

electrodes showing significantly greater HFB contribution of external displacements compared to saccades and one V2 electrode showing the opposite effect. Last, comparing the saccades with only the small (1.3°, *Figure 2—figure supplement 2*) or large (3.7°, *Figure 2—figure supplement 3*) external displacements also found qualitatively similar results, yet with a lower number of significant sites (25/115 and 27/115 significant electrodes, respectively), likely due to the reduced statistical power of these smaller sample analyses.

## External displacements vs. saccades response latencies

Examining the sites that showed significant HFB increases to both external displacements and saccades (according to a criterion of $p_{FDR} < 0.05$, corrected within patient), we timed the latencies of the peaks of the HFB increases driven by each of these two events. Prior to peak detection, the responses were divided by their standard error in order to attenuate noise-induced peaks. We found a significant linear correlation ($r = 0.66$, $n = 34$, p=0.00002) between the peak response latency of external displacements and the equivalent measure for saccades across electrodes. There was no significant difference in the average latencies of these two conditions (a paired t-test across electrodes, $t(33) = 1.15$, p=0.26, $M = 173.3 \pm 51.0$ ms for external displacements, $M = 164.8 \pm 53.2$ ms for saccades). These results are consistent with a retinal origin of the HFB increases observed following both external displacements and saccades. However, it is important to emphasize these results were mainly observed in early visual sites, while in higher level regions the response to saccades was suppressed (precluding reliable latency measurement of the residual response).

## Single cases with simultaneous low- and high-level recordings

Since these results were pooled across multiple subjects, one might argue that the observed pattern of increased differential response to displacements versus saccades along the cortical hierarchy might be an artifact produced by the juxtaposition of effects recorded in different individuals, who might be confounded by uncontrolled behavioral or neuronal differences. Evidence against this possibility is provided by saccade and displacement-related responses acquired simultaneously in low and high-level sites. This is illustrated in *Figure 3* in which each panel depicts recordings obtained in different electrodes implanted within an individual subject. The higher-level traces demonstrate marked HFB activity increases following external displacements but not following saccades. The traces from early visual sites show similar HFB response increases following both events.

Some additional, more subtle aspects of the response to displacements versus saccades are evident in these individual traces. In subject P57 (*Figure 3a*), V1 (left column) and MST (right column, an electrode located slightly less than 1 cm anterior to the anatomically defined MST/TO2) sites were recorded simultaneously. In this particular case, the saccade-related activity increase in V1 was

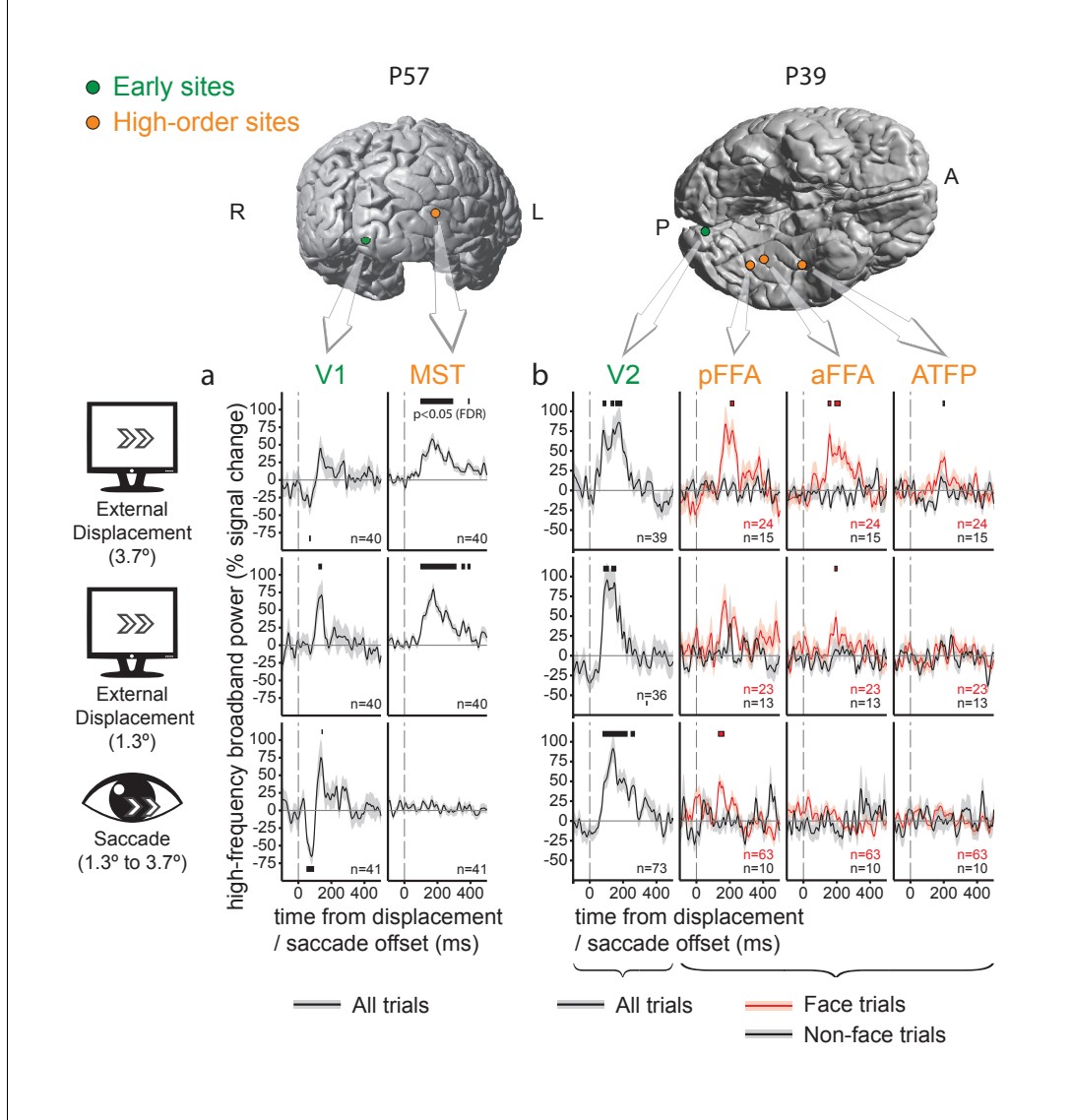

**Figure 3.** Individual electrode traces of the contribution of external stimulus displacements and saccades to the High-Frequency Broadband activity in low vs. high-level sites, sampled in two individual subjects. Shading around the waveforms denotes ±1 standard error of the regression coefficients (between-trials error variability). Horizontal bars mark response timepoints significantly different from zero (p<0.05, FDR-corrected within-participant). The number of event occurrences is denoted by n. Electrode locations are marked on the subjects' individual FreeSurfer-reconstructed cortices. (a) V1 versus MST. Note the insensitivity of the MST site to saccades. (b) V2 versus three face-selective sites. Note that saccade-sensitivity decreases first along the hierarchy, followed by a decrease in displacement sensitivity at more anterior sites.

DOI: https://doi.org/10.7554/eLife.27819.009

preceded by an activity decrease. This trend was not sufficiently strong to pass our false-discovery rate control when compared with displacements at the group level.

In subject P39 (*Figure 3b*), a V2 response (leftmost column, responses collapsed over face and non-face trials) is compared with three face-selective sites sampling posterior and anterior FFA (FFA-1 and FFA-2,*Weiner and Grill-Spector, 2012*) and a probable anterior temporal face patch (rightmost column, see *Rajimehr et al., 2009*). Beyond replicating the pattern of HFB activity increase in high-level visual regions following external displacements but not after saccades, this particular case may hint at a hierarchical gradient within face-selective sites, with insensitivity to the saccadic displacement at anterior but not posterior FFA and relatively invariant response both to saccades and external displacements in the anterior face patch site.

## Comparison of suppression of eye blink- and saccade-related transients

An interesting question concerns the possible relationship between the suppression of saccadic-related transients and the previously reported (*Golan et al., 2016*) suppression of blink-related transients. To quantitatively test for such an association, we first examined the level of overlap between two sets of electrodes: those that showed a significantly greater HFB increase in response to external displacements compared with saccades and those that showed a significantly greater HFB increase in response to 'gaps' (blank frames) compared with eye blinks obtained from *Golan et al. (2016)*. The overlap between the two sets of electrodes was far above chance level, with a Sørensen–Dice coefficient of 0.5763 (p=0.000018, random permutation test, see Materials and methods).

In order to gain a more detailed understanding of the association between the saccade and blink suppression effects, we defined two indices as follows: First, we defined a suppression index for blinks, $\frac{gap-blink}{gap+blink}$, where 'gap' indicates the HFB increase following the offset of blank frames, while 'blink' indicates the HFB increase following the offset of spontaneous blinks. For this measure we used the cross-validated response estimates for both events (see *Golan et al., 2016*). Similarly, we defined a suppression index for saccades as $\frac{displacement-saccade}{displacement+saccade}$. *Figure 4* depicts the two resulting indices, averaged within each of the seven ROIs. All of the ROIs except for V4 were well fit by a linear relation between the two indices (*r* = 0.95, *n* = 6). Since such an opportunistic removal of an outlier introduces positive bias to the correlation, the correlation's significance was tested by a randomization procedure that takes into account this issue (p=0.0111, see Materials and methods). V4 was notably above the trend line, showing greater suppression of saccades than expected. However, since our V4 sample was small (6 electrodes), we cannot unequivocally determine from the current

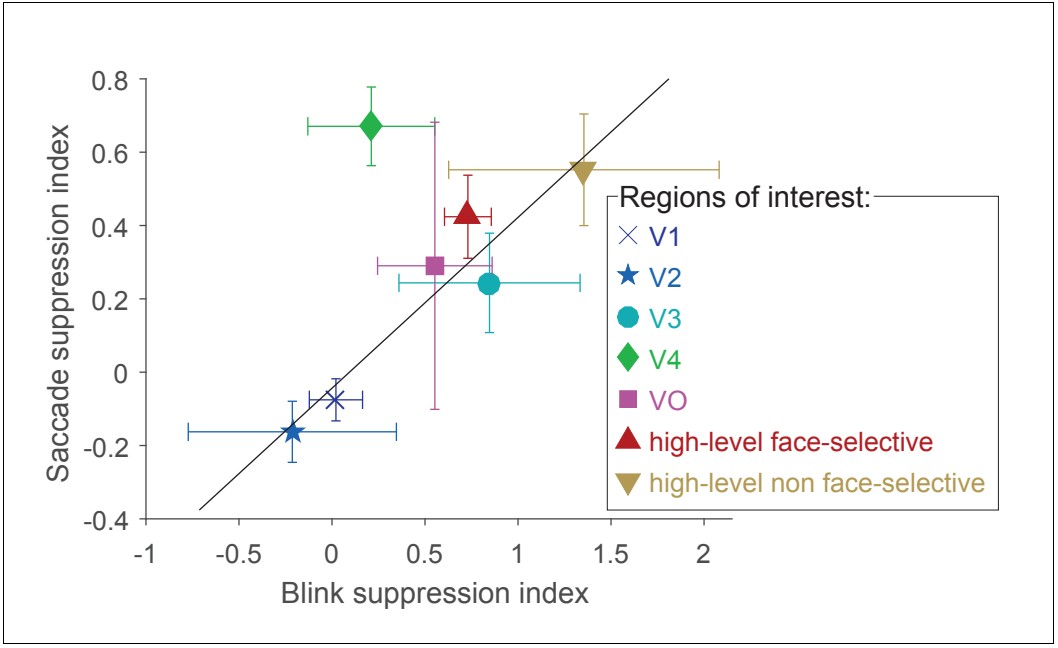

**Figure 4.** Blink suppression index $\left(\frac{gap-blink}{gap+blink}\right)$ versus saccade suppression index $\left(\frac{displacement-saccade}{displacement+saccade}\right)$ across seven visual regions of interest. The circles depict the ROIs' means and the error bars depict the standard errors of the means. Values greater than 1 were possible since some electrodes registered negative blink- (or saccade-) related responses. The trend line (in black, *r* = 0.95) was fit to all regions except V4 (see text for bias-corrected testing of this fit).
DOI: https://doi.org/10.7554/eLife.27819.010

The following source data is available for figure 4:

**Source data 1.** Individual electrode data for *Figure 4*.
DOI: https://doi.org/10.7554/eLife.27819.011

data whether this sample-level deviation reflects that V4, in general, has a particularly strong suppression effect for saccades compared with blinks.

## Discussion

We compared human intracranial recordings following retinal displacements of naturalistic stimuli caused either by small external displacements of the stimuli or by saccades of similar magnitude. External displacements triggered a subsequent HFB activity increase that affected both lower-level visual regions and higher-level visual regions, including both the dorsal and ventral streams. In contrast, the activity increase following saccades was mostly confined to early visual areas—V1 to V3—and waned to nil as the visual signal proceeded forward in the visual hierarchy. This spatiotemporal response pattern was reflected in the gradual appearance of significant differences between the external displacement-related and saccade-related HFB activity increases along the progression of the cortical hierarchy.

### Retinal versus extra-retinal accounts

Do the observed differences between displacements and saccades in high-level visual cortex indicate an active (i.e., extra-retinal) suppression of the saccade-related responses or a passive product of low-level differences in the retinal stimulation between the two conditions? Due to limitations imposed by the bedside setting, the external displacements of the stimuli did not follow a precise replay of the subjects' tracked saccades (as in, e.g., *Troncoso et al., 2015*). Whereas we matched the magnitudes of the saccades and the displacements by post hoc selection, there were remaining differences in direction, latency, and gaze location. Nevertheless, with regard to the primary effect we report, which is the appearance of HFB increases in higher-level visual cortex following external displacements but not saccades, we believe a retinal account of the effect is unlikely. Similar to the case of blink-related responses (*Golan et al., 2016*), low-level retinal discrepancies between saccades and displacements can be expected to have the largest effect on neuronal responses in early stages of the cortical hierarchy – that are particularly sensitive to such low-level parameters – and a smaller effect on high order processing (see *Grill-Spector and Malach, 2004*). However, our results show the opposite trend, with similar responses to displacements and saccades in V1 and increasing discrepancies between saccades and displacements at higher levels of the hierarchy.

It is less certain whether the weak reductions in HFB activity following saccades, observed in some early foveal visual sites (e.g. *Figure 3a*), are necessarily driven by an extra-retinal signal. There are multiple findings of modulation of the early visual cortex activity by saccades in total darkness (e.g., *Sylvester et al., 2005*) and even during REM sleep (*Uematsu et al., 2013*), providing evidence that at least some extra-retinal saccade-related signal arrives at the early visual cortex. However, retinal contributions could also explain the observed activity reductions. Since the external displacements we employed were simply an update of the stimulus location on an LCD screen, they did not simulate the momentary blurring of the entire visual field caused by saccades. Obviously, such a momentary reduction in spatial contrast is a plausible contributor to temporary HFB reductions in early visual areas. This retinal account of saccade-related activity decrease may also apply in other electrophysiological studies that reported saccade-related reductions of activity in early visual cortex while relying on displacing stimuli on a computer monitor as a retinal control (e.g., *Troncoso et al., 2015*). Elucidating the relative contribution of retinal and extra-retinal effects in early visual cortex will require further electrophysiological studies employing a finer simulation of the retinal impact of saccades, such as by using a rotating mirror, as was done in certain psychophysical studies (*Diamond et al., 2000*).

### Visual stability actively develops along the visual hierarchy

The finding of diminishing HFB responses to saccades compared to external displacements along the visual hierarchy is compatible with the suppression of saccade-related responses previously observed in non-human primate MT/MST (*Thiele et al., 2002*). However, the current findings extend earlier reports in two respects. First, the concurrent recordings across the visual hierarchy reveal a striking contrast between lower and higher-level cortical regions in representing self-generated versus external stimulus displacements. In particular, the similar HFB increases in early visual cortex following external displacements and saccades indicate that despite the documented extra-retinal

effects in V1, this region's activity is strongly perturbed by small saccades, compared with higher-level visual cortex. As in the case of blinks (*Golan et al., 2016*), the 'unstabilized' nature of early visual cortex provides further evidence that V1 activity is incompatible with the individual's subjective perceptual awareness (as hypothesized by *Crick and Koch, 1995*).

A second advance of the current results is the extension of the observed saccadic suppression effect to ventral high-level visual cortical regions, including face-selective sites. The results further confirm that in contrast to the traditional view of the category-selective ventral visual cortex as largely position and motion invariant, these areas clearly show a considerable sensitivity to small external displacements. Furthermore, the finding that these transient responses to displacement are quenched during saccades indicates that extra-retinal saccade-related information affects not only the dorsal visual processing stream but also the ventral one. This result is compatible with the suggestion of an extra-retinal stabilization mechanism in the ventral stream (*Leopold and Logothetis, 1998*). Our results support this conclusion by directly comparing external displacements and saccades, thus precluding the alternative of a purely passive stabilization achieved through large receptive fields.

### Limitations and open questions

In the present study, we focused on saccades considerably smaller than the displayed object (see Materials and methods). When saccades are larger, they are likely to bring new objects into the fovea and thus change the high-level neural responses (*DiCarlo and Maunsell, 2000*; *Hamamé et al., 2014*; *Podvalny et al., 2017*).

Another reservation is that the displacement versus saccade contrast does not effectively assess visual regions that are insensitive to both. Such regions may lie at the more anterior end of the ventral high-level visual cortex (e.g. *Figure 3b*, rightmost column).

Last, it has to be noted that our results are limited to HFB activity: While it is the best iEEG correlate of average spiking rate (*Mukamel et al., 2005*), it is possible that single unit activity (which was not recorded in our subjects) or slower iEEG frequencies (which we did not analyze due to the limited resolvability of their spatial sources) might show different result patterns.

### Indication for a shared/overlapping pathway for suppression of eye blink- and saccade-related transients

Finally, our analysis indicates a strong resemblance in how saccade-related and blink-related transients are suppressed along the cortical hierarchy (See Results and *Figure 4*). Along with the psychophysical resemblance between blink and saccadic suppression (*Ridder and Tomlinson, 1997*), these findings suggest that motor-visual pathways informing higher-level visual cortex of blinks and saccades considerably overlap. An interesting possibility is that these two oculomotor events are utilizing a single, shared descending pathway. The latter hypothesis is supported by the systematic occurrence of small transient downward-nasalward eye movements during eye blinks (*Collewijn et al., 1985*). Furthermore, it is intriguing to note that from an evolutionary perspective (kindly suggested by one of the anonymous reviewers), the pressure to suppress the impact of saccades and eye retractions on vision was exerted already on our eyelidless aquatic ancestors. Hence, when the need to suppress the visual impact of eye blinks first appeared, a pathway linking extra-ocular activity to suppression of the visual response was probably already present.

However, the present results leave open the question of the precise mechanism underlying the saccade-related suppression of activity transients in ventral stream representations. That mechanism may be a general gating mechanism—suppressing all motion-related transients regardless of saccade direction—or a more specific offsetting by oculomotor motion vectors, similar to that previously described in monkey MT (*Thiele et al., 2002*). Future experiments, e.g. in which the visual image is displaced concurrently with the saccade, will be needed to resolve this issue.

## Materials and methods

Unless stated otherwise, all data acquisition and analysis steps are compatible with the Materials and methods section of *Golan et al. (2016)*, including electrode-localization, regions of interest (ROIs) definition, data modeling and statistical testing.

## Subjects

The data reported here is a product of a reanalysis of the experimental sessions reported in *Golan et al. (2016)*, except for a single subject who was tested later on (patient code P68, 29 Y/O, female, right hemisphere Lateral Occipital and Fusiform Gyrus seizure onset zones, SOZs excluded from analysis) and three subjects who were excluded from the current report either due to lack of video eye-tracking (P20 and P25) or due to lack of a sufficient number of matching saccades (P46). Thus, this report includes data recorded from 12 patients. All patients gave fully informed consent, including consent to publish, according to NIH guidelines, as monitored by the institutional review board at the Feinstein Institute for Medical Research, in accordance with the Declaration of Helsinki.

## Eye position tracking and analysis

Eye position was tracked by an EyeLink 1000 video eye tracker (EyeLink 1000, SR Research, Ontario, Canada), operating at 500 Hz (0.05° RMS resolution). Eye tracking was obtained without head-stabilization, using a head-tracking enabled by a forehead sticker. However, in practice, only little head movement occurred since the patients rested their head on a pillow. A standard 9-point calibration was followed by a validation procedure that ensured tracking quality was estimated as 'GOOD' by the eye tracker's acquisition software (no more than 0.5° average error and 1° maximal error across fixation targets). When necessary, calibration was repeated after the break between the two experimental runs.

Periods of no successful tracking or where eye position was estimated to lie outside the stimulus borders were discarded from analysis, as well as the preceding 50 ms and following 250 ms (eye blinks were excluded from this rejection criterion). Saccades were detected using the built-in saccade detection algorithm of EyeLink 1000, excluding saccade events encompassing blink events. Saccade magnitudes were converted from pixels to visual angle based on screen distance (70 cm) and screen pixel density (78.5 DPI). For example, the small (50 pixels) and large (140 pixels) displacements were translated into 1.3 and 3.7 visual degrees.

## Modeling of saccades and displacements

As in *Golan et al. (2016)* a Finite Impulse Response (FIR) General Linear Model enabled estimating the contribution of each experimental event to the HFB timecourse of each electrode while correcting for overlap artifacts. Saccades and displacements were modeled by FIR predictor sets, each set spanning from 300 ms before displacement/saccade offset to 500 ms following displacement/saccade offset. Using saccades' onsets instead of offsets produced very similar results. Saccades smaller than 1.3° or larger than 3.7° were modeled by two additional separate predictor sets. External displacements were either modeled separately for different magnitudes (*Figures 1* and *3*) or by a single predictor set (*Figure 2*). For *Figures 1* and *3*, saccades and displacements were estimated separately for face and non-face events. New stimulus appearance (face and non-face stimuli), gaps and blinks were modeled as in *Golan et al. (2016)*.

## Sørensen–Dice coefficient

The Sørensen–Dice coefficient is a binary association measure that is defined as $\frac{2|A \cap B|}{|A|+|B|}$, where *A* and *B* in our particular case were the sets of significant electrodes in Figure 2 here and in Figure 6 of *Golan et al. (2016)*, respectively. For this measure, we used only the 108 visually-responsive electrodes obtained from the 11 patients who were included in both this report and in *Golan et al. (2016)*. Statistical testing of the resulting index against chance was done by randomly permuting (100,000 random permutations) the 108 electrode labels within one of the statistical maps.

## Statistical testing for correlation with an outlier removed

Since we computed a correlation after removing a notable outlier, standard statistical testing of the correlation coefficient would be circular and invalid. Therefore, we tested the correlation against chance using a bias-corrected randomization procedure. Simulating the null hypothesis of no correlation by shuffling the datapoints (*Howell, 2015*), we repeatedly measured the correlation obtained after removing the data point that most hindered the correlation coefficient in each simulation (i.e., taking the maximal correlation value across all possible single-outlier removals). This resulted in a

null distribution of the biased correlation coefficient (which was, in our case, centered on 0.29 instead of 0.00). This distribution was then used to calculate a valid $p$ value.

## Acknowledgements

We thank the anonymous reviewers for their constructive comments. We are grateful to the participating patients, who contributed their time and effort to this study.

## Additional information

### Competing interests

Charles E Schroeder: Reviewing editor, *eLife*. The other authors declare that no competing interests exist.

### Funding

| Funder | Grant reference number | Author |
|---|---|---|
| Canadian Institute for Advanced Research | Azrieli Program in Brain Mind and Consciousness | Rafael Malach |
| US-Israel Binational Science Foundation | 2013-070-2 | Leon Y Deouell |
| US-Israel Binational Science Foundation | Rahamimoff Travel Grant for Young Scientists. T-2014215 | Tal Golan |
| European Commission | Marie Curie International Outgoing Fellowship within the 7th European Community Framework Programme | Lucia Melloni |

The funders had no role in study design, data collection and interpretation, or the decision to submit the work for publication.

### Author contributions

Tal Golan, Conceptualization, Formal analysis, Methodology, Writing—original draft, Acquisition of data; Ido Davidesco, Conceptualization, Writing—review and editing, Acquisition of data; Meir Meshulam, David M Groppe, Pierre Mégevand, Erin M Yeagle, Matthew S Goldfinger, Writing—review and editing, Acquisition of data; Michal Harel, Implant digital reconstruction; Lucia Melloni, Methodology, Writing—review and editing; Charles E Schroeder, Leon Y Deouell, Ashesh D Mehta, Rafael Malach, Conceptualization, Writing—review and editing

### Author ORCIDs

Tal Golan, http://orcid.org/0000-0002-7940-7473
Ido Davidesco, http://orcid.org/0000-0003-0754-5807
Meir Meshulam, http://orcid.org/0000-0001-5899-7681
David M Groppe, http://orcid.org/0000-0002-3282-2514
Pierre Mégevand, http://orcid.org/0000-0002-0427-547X
Erin M Yeagle, http://orcid.org/0000-0003-1147-4976
Matthew S Goldfinger, http://orcid.org/0000-0001-9274-8742
Lucia Melloni, http://orcid.org/0000-0001-8743-5071
Leon Y Deouell, http://orcid.org/0000-0002-6147-5208
Ashesh D Mehta, http://orcid.org/0000-0001-7293-1101
Rafael Malach, http://orcid.org/0000-0002-2869-680X

### Ethics

Human subjects: Human subjects: All patients gave fully informed consent, including consent to publish, according to NIH guidelines, as monitored by the institutional review board at the Feinstein

Institute for Medical Research, in accordance with the Declaration of Helsinki. Data was obtained as part of protocol number 07-125. Patients had the opportunity to consent prior to electrode implantation and were informed that they may choose to decline or later withdraw from the study without affecting their clinical care. Consent includes agreement to participate with studies of cognitive and sensorimotor processes and publication of any deidentified data obtained. Risks include tedium and potential breach of medical information and are minimized by giving ample breaks and implementation of protocols to deidentify data close to the time of recording. Benefits to the subject include increased monitoring of the electrocorticogram and involvement of research methods to help localize electrodes with respect to preoperative MRI.

### Decision letter and Author response
Decision letter https://doi.org/10.7554/eLife.27819.016
Author response https://doi.org/10.7554/eLife.27819.017

## Additional files

### Supplementary files
• Transparent reporting form
DOI: https://doi.org/10.7554/eLife.27819.012

### Major datasets
The following previously published dataset was used:

| Author(s) | Year | Dataset title | Dataset URL | Database, license, and accessibility information |
|---|---|---|---|---|
| Wang L, Mruczek RE, Arcaro MJ, Kastner S | 2015 | Probabilistic Maps of Visual Topography in Human Cortex | http://scholar.princeton.edu/napl/resources | Publicly available at Princeton University Website |

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
