## [Decision Letter]

Thank you for submitting your article "Increasing suppression of saccade-related transients along the human visual hierarchy" for consideration by *eLife*. Your article has been reviewed by two peer reviewers, and the evaluation has been overseen by a Reviewing Editor and David Van Essen as the Senior Editor. The reviewers have opted to remain anonymous.

The reviewers have discussed the reviews with one another and the Reviewing Editor has drafted this decision to help you prepare a revised submission.

Summary:

Your manuscript submitted as a Research Advance, a follow up on your recently published paper in *eLife,* was well received by the reviewers, who nevertheless raised two issues that I hope you will address in your revision.

Essential revisions:

One concerns additional analysis of temporal dynamics and latencies of neural responses with respect to stimulus displacement to address the relationship between blink and saccade effects across cortical areas. The second, relatively minor issue, concerns the suggested overlap between visuomotor pathways that signal blinks and saccades to higher-level visual cortical areas. In the Discussion, please consider a possibility of the operation of a single extraocular pathway for suppression, as suggested by one of the reviewers.

Reviewer #1:

The present study investigated activity evoked by small saccades at different stages of visual cortical processing in human patients implanted with ECoG electrodes. It uses the same data set from a previous study, which focused on the responses to blinks. As in the previous study, the authors focused on high frequency LFP signals, which have the advantage of being both spatially and temporally resolved, and often serves as a reasonable proxy for local spiking.

In the present study, the responses during saccades are compared to responses to physical stimulus displacement (during fixation). The data come from 12 patients, in which 115 contacts showed visual responses, according to the approximate assignment of visual areas, based on methods described in Golan et al., 2016. Through this splitting, the number of electrodes per area ranged between 7 and 12 electrodes per area. In two of the patients (Figure 3) there were simultaneous recordings from early retinotopic areas (V1, V2) and higher-level areas (MST, FFA), allowing for comparison between low- and high-level visual areas over the same trials.

The featured effect is a pronounced difference between low- and high-level visual areas in how they respond during small saccades. Whereas most visually responsive areas were sensitive to small (1.3-3.7 degree) physical displacements of the image during fixation, only the early visual cortex sites responded with equivalent strength during corresponding amplitude eye movements. Beyond the retinotopic areas in both dorsal and ventral streams, saccades elicited relatively little perturbation of neural activity. This difference between physical displacement and saccades can be seen in Figure 1, and the difference across areas is shown spatially in Figure 2, where the red dots indicate significantly higher responses for physical displacement than for small saccades of similar amplitude.

In some ways, the outcome of this study is consistent with the prevailing thinking of saccadic responses across the visual cortex. Previous work has shown that saccades have stronger effects in retinotopic than higher-level visual areas, where such effects in the dorsal stream have been linked with extraretinal inputs and saccadic suppression. An interesting finding in the present study was that small physical displacements led to such strong responses in high-level visual cortex (e.g. in the face-selective electrodes in Figure 1), which was for me a bit unexpected. It argues that the observed lack of responses (or diminished responses) during saccades cannot easily be explained by something like receptive field size alone. However, the authors are circumspect in describing this, and concede that to pin down the retinal or extraretinal nature of such results would require a more thorough study.

Overall, I find the study to be a valuable contribution, particularly since it compares multiple areas in the human brain. There are a few aspects that I think could be improved. For example, more analysis could be done on the temporal dynamics of the neural responses, including the response latencies relative to the stimulus displacement and saccadic landing, and how this changes across areas. I also found myself wondering about the correspondence of the sites that are sensitive/insensitive to saccades and those related to blinks from the previous study. The authors may see that as out of the scope of the current paper, but they might consider adding analysis to show whether there is any relationship between the blink effects and saccade effects across electrodes.

Reviewer #2:

In this manuscript, the investigators add analysis of saccadic suppression to their previous study of blink suppression (Golan et al., 2016). Based on the authors' revision of the previous manuscript on blinking, I have no technical or analytical issues with the current manuscript. In the Discussion comparing blink and saccadic suppression, I think, however, that the authors might give more consideration to the issue of eye movements associated with blinking.

The authors write (subsection “Indication for overlapping pathways for suppression of eye blink- and saccade-related transients”): "Taken together with the psychophysical resemblance between blink and saccadic suppression (Bidder II and Tomlinson, 1997), these findings suggest overlapping motor-visual pathways informing higher-level visual cortex of both blinks and saccades." I think that a more likely possibility is that it is a single motor-visual pathway for suppression, rather than overlapping, pathways. This interpretation follows from the fact that the extraocular system creates both saccadic eye movements and the eye movements with blinks. In humans, co-contraction of multiple extraocular muscles to create eye retraction causes the rotational eye movements associated with a blink. Extraocular muscle co-contraction associated with blinking almost certainly developed before eyelids. Beginning with terrestrial species, the eyelid's primary role was to spread the tear film across the cornea to maintain corneal moisture. Aquatic species lack eyelids because corneal moisture is not an issue, but they exhibit eyeball retraction as a protective mechanism for the cornea. We focus on vision being obscured as the critical reason for blink suppression. Nevertheless, complete closure of the eyelids is rarely more than half of the blinks we make each day, while the eye movements with a blink occur with every blink. It seems more likely that our early ancestors developed extraocular system suppression to deal with saccadic rotational eye movements and the rapid eye retraction of 'blinking'.

---

## [Author Response]

*Reviewer #2:*

*[…] Overall, I find the study to be a valuable contribution, particularly since it compares multiple areas in the human brain. There are a few aspects that I think could be improved. For example, more analysis could be done on the temporal dynamics of the neural responses, including the response latencies relative to the stimulus displacement and saccadic landing, and how this changes across areas.*

We thank the reviewer for his/her favorable opinion of the manuscript and for these suggestions. We fully agree that a more detailed analysis of the potential similarities/differences in the response latencies to external displacements vs. saccadic landings could be a valuable addition to the manuscript. However, such an analysis would have to be confined to sites in which the response to saccadic landing was reliable enough to allow latency measurements. As we show in the results, in most high-level visual sites, the responses related to saccadic-landings were greatly diminished (and often, completely absent), consequently rendering the measurement of saccadic response latencies unattainable. Therefore, we have examined the relationship between the latencies of the responses in sites that responded to both events. We report the results of this analysis in the Results section (“External displacements vs. saccades response latencies”).

*I also found myself wondering about the correspondence of the sites that are sensitive/insensitive to saccades and those related to blinks from the previous study. The authors may see that as out of the scope of the current paper, but they might consider adding analysis to show whether there is any relationship between the blink effects and saccade effects across electrodes.*

We thank the reviewer for pointing to this interesting possible generalization. Following this comment, we quantitatively analyzed the relation between the suppression effects of blinks and saccades and introduced a new subsection in the Results section. We chose to focus on the normalized difference between gaps and blinks compared with the normalized difference between external displacements and saccades. Unlike raw blink- and saccade- related responses, these normalized measures are controlled for potentially confounding overall visual responsivity, a factor that would otherwise generate strong but irrelevant associations. We report on this new analysis in the Materials and methods section (“Sørensen–Dice coefficient” and “Statistical testing for correlation with an outlier removed”) and in the Results section (“Comparison of suppression of eye blink- and saccade-related transients”). The relevant Discussion section addition also relates to reviewer 2 concerns and can be found in the response to his/her comments.

*Reviewer #2:*

*In this manuscript, the investigators add analysis of saccadic suppression to their previous study of blink suppression (Golan et al., 2016). Based on the authors' revision of the previous manuscript on blinking, I have no technical or analytical issues with the current manuscript. In the Discussion comparing blink and saccadic suppression, I think, however, that the authors might give more consideration to the issue of eye movements associated with blinking.*

*The authors write (subsection “Indication for overlapping pathways for suppression of eye blink- and saccade-related transients”): "Taken together with the psychophysical resemblance between blink and saccadic suppression (Bidder II and Tomlinson, 1997), these findings suggest overlapping motor-visual pathways informing higher-level visual cortex of both blinks and saccades." I think that a more likely possibility is that it is a single motor-visual pathway for suppression, rather than overlapping, pathways. This interpretation follows from the fact that the extraocular system creates both saccadic eye movements and the eye movements with blinks. In humans, co-contraction of multiple extraocular muscles to create eye retraction causes the rotational eye movements associated with a blink. Extraocular muscle co-contraction associated with blinking almost certainly developed before eyelids. Beginning with terrestrial species, the eyelid's primary role was to spread the tear film across the cornea to maintain corneal moisture. Aquatic species lack eyelids because corneal moisture is not an issue, but they exhibit eyeball retraction as a protective mechanism for the cornea. We focus on vision being obscured as the critical reason for blink suppression. Nevertheless, complete closure of the eyelids is rarely more than half of the blinks we make each day, while the eye movements with a blink occur with every blink. It seems more likely that our early ancestors developed extraocular system suppression to deal with saccadic rotational eye movements and the rapid eye retraction of 'blinking'.*

We thank the reviewer for raising the intriguing possibility of a unified pathway underlying both blinks and saccades. We have edited and expanded the relevant discussion paragraph to better account for this hypothesis. We further thank the reviewer for pointing to the possible evolutionary link between saccade related and blink related suppression effects. We have included a reference to this insightful idea in the revised Discussion section (“Indication for a shared/overlapping pathway for suppression of eye blink- and saccade-related transients”) and acknowledged the reviewer's contribution.